# UNSUPERVISED MULTI-SENSOR SPECTRAL IMAGE FUSION VIA FREQUENCY-SPATIAL RECIPROCAL-VIEW LEARNING

## ABSTRACT

The unsupervised fusion of multi-sensor spectral images is often limited by non-absolute registration. This misalignment leads to significant differences between the spectral shape of the fused image and the original hyperspectral signal. To address this challenge, we propose the Frequency-Spatial Reciprocal-View Learning (FSRVL) for unsupervised multi-sensor MSI-HSI fusion. 1) Feature Synchronization: weight-sharing convolutions are employed to process Low-Resolution Hyperspectral Image (LR-HSI) and High-Resolution Multispectral Image (HR-MSI) in the frequency domain jointly, achieving an information correspondence between the two modalities with parameter transfer. 2) Frequency Recalibration: sub-pixel information is assigned to learnable filters to adaptively refine spatial distributions across various materials and promote the reestablishment of their spectral characteristics. The advantages of FSRVL were demonstrated across various simulated and real-world scenarios, with experiments confirming that FSRVL outperforms the baselines. *The source code will be linked here.*

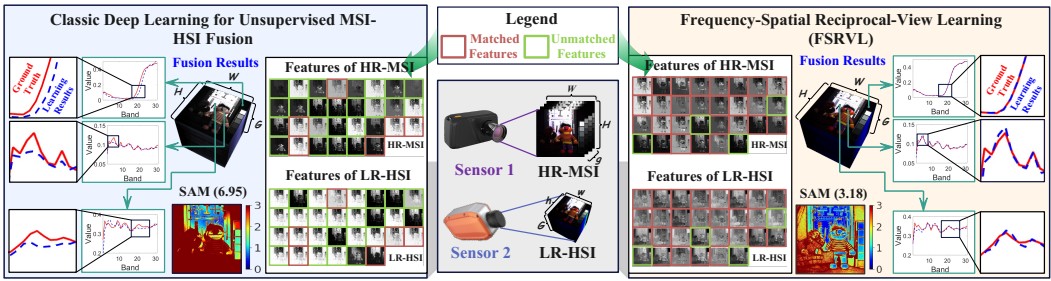

Figure 1: When using traditional deep learning methods for unsupervised MSI-HSI fusion, it typically encounters the information mismatch between sensors. Our FSRVL can effectively increase matched features to mitigate this issue.

## 1 INTRODUCTION

The unsupervised fusion of Hyperspectral Image (HSI) and Multispectral Image (MSI) is typically guided by spectral unmixing, wherein the Low-Resolution Hyperspectral Image (LR-HSI) serves as the source of Endmember Signal Dictionary (ESD), and the High-Resolution Multispectral Image (HR-MSI) provides the Corresponding Subpixel Abundances (CSAs) (Yang et al., 2024a). A High-Resolution Hyperspectral Image (HR-HSI) is then reconstructed by combining the two (Xie et al., 2019). Despite the success of both deep learning-based and handcrafted methods, their practicality is fundamentally limited by the assumption that ideal correspondence exists between LR-HSI's ESD and HR-MSI's CSAs in the spatial domain. This assumption is rarely satisfied in multi-source scenarios, severely hindering the quality and reliability of the fusion data (Qu et al., 2025).

Figure 1 highlights the key bottleneck in current deep learning-based unsupervised MSI-HSI fusion, where most methods rely on the ideal assumption that endmember signals and abundance fractions

from HSI and MSI correspond perfectly. For example, the $i$-th endmember signal in the ESD of the HSI is assumed to represent the same material as the $i$-th endmember signal in the ESD of MSI, and that the abundance vectors obtained from both datasets are also corresponding. However, in practice, the MSI and HSI do not always correspond ideally, which might lead to low-quality fusion performance (Dong et al., 2024). Therefore, we develop the Frequency-Spatial Reciprocal-View Learning (FSRVL) to address the issue of information mismatch for unsupervised MSI-HSI fusion. Figure 1 adopts a CNN to acquire the results of classic deep learning.

HSIs are data cubes that contain dozens to hundreds of spectral bands (Zhang et al., 2024). Unlike other types of images, HSIs provide more detailed spectral data, which can reflect the intrinsic characteristics of various objects (Yang et al., 2024b). However, due to the limitations of imaging hardware, HSIs have to face a trade-off between spectral resolution and spatial resolution (Zeng et al., 2025; Wang et al., 2024). Specifically, increasing the spectral resolution leads to a decrease in spatial resolution, while enhancing spatial resolution causes a reduction in spectral resolution (Su et al., 2025). To improve the spatial resolution of a single image, one must either use more advanced sensors or process the image by supervised learning (Li et al., 2024). The former comes with enormous costs, while the latter relies on prior knowledge and some degree of manual expertise (Zhang et al., 2020; Qu et al., 2024). Conversely, MSIs have fewer spectral channels compared to HSIs, but offer higher spatial resolution and are generally more cost-effective to obtain (Zhu et al., 2025).

Nowadays, the fusion of an LR-HSI and an HR-MSI can reconstruct a new HR-HSI, which can achieve HSI super-resolution (Liu et al., 2022; Gao et al., 2023). In response to this demand, numerous HSI-MSI fusion methods have been developed, such as DFMFf (Guo et al., 2023), SSG-SRL (Liu et al., 2025), and OTIAS (Deng et al., 2025). However, these methods are supervised, meaning their fusion performance heavily relies on manual sampling, limiting their scalability. Conversely, unsupervised HSI-MSI fusion is more capable of meeting the user demand for high-resolution HSI (HR-HSI). This has led to the development of several representative methods, including CUCaNet (Yao et al., 2020), DTDNML (Wang et al., 2025), and EDIP (Li et al., 2025). Currently, most unsupervised fusion approaches assume that the MSI and HSI imaging records under the same scene are identical, and they use spectral mixing models as the linkage between the two modalities. This manner preserves the spectral information from LR-HSI and the spatial details of HR-MSI. Note that this requires a perfect correspondence between the CSAs of MSI and the ESD of HSI. However, this ideal registration assumption is problematic to meet in practical multi-source data fusion, as distortions cannot be fully eliminated, inevitably leading to issues of information mismatch between the modalities.

To overcome the technical bottleneck for unsupervised multi-sensor fusion, we design a new approach, named FSRVL. This novel deep learning framework transforms MSI and HSI into the frequency domain, enabling adaptive fusion by applying frequency filtering. Specifically, we adapt the discrete Fourier transform to process CSAs corresponding to ESD for both MSI and HSI modalities, where spatial details are encoded in the phase components and global structure is encoded in the amplitude. FSRVL can adaptively harmonize the CSAs in hidden layers for both MSI and HSI modalities, thereby improving information matching. ESDs of the two modes correspond to the weights of abundance features, and they will also become increasingly aligned during learning. Although this serves as an illustration, all pictures are derived from real experiments. As shown in Figure 1, FSRVL significantly increases the number of information matches between LR-HSI and HR-MSI, thereby substantially enhancing the fusion performance.

The main contributions of this paper are summarized as:

- FSRVL introduces the idea of deep frequency filtering to address the misalignment of modal information within the same scene during unsupervised MSI-HSI fusion, thereby enhancing performance.

- We analyze the issue of mismatch between the Endmember Signal Dictionary (ESD) of LR-HSI and the Corresponding Sub-pixel Abundances (CSAs) of HR-MSI, providing a novel direction for enhancing the effectiveness of unsupervised MSI-HSI fusion.

- This paper not only used a simulated scenario with Ground Truth to quantitatively evaluate FSRVL, but also validated it using unregistered multi-source datasets.

## 2 RELATED WORKS

### 2.1 UNSUPERVISED MSI-HSI FUSION

This paper defines the HR-MSI and LR-HSI using the notations $\boldsymbol{X} \in \mathbb{R}^{WH \times g}$ and $\boldsymbol{Y} \in \mathbb{R}^{wh \times G}$, respectively. Here, $(W, H)$ and $(w, h)$ represent the spatial sizes of the HR-MSI and LR-HSI datasets, and $g$ and $G$ denote the number of bands corresponding to the HR-MSI and LR-HSI, respectively. Generally, they also should have the following relationship: $W \gg w$, $H \gg h$, and $G \gg g$ (Li et al., 2025). When the Linear Spectral Mixing Model (LSMM) is employed, HR-MSI and LR-HSI can be expressed as:

$$\boldsymbol{X} = \boldsymbol{A}\widehat{\boldsymbol{E}}, \quad \boldsymbol{Y} = \widehat{\boldsymbol{A}}\boldsymbol{E} \tag{1}$$

where $\widehat{\boldsymbol{E}} \in \mathbb{R}^{e \times g}$ and $\boldsymbol{E} \in \mathbb{R}^{e \times G}$ denote ESDs of $\boldsymbol{X}$ and $\boldsymbol{Y}$, respectively. In Eq. (1), $\boldsymbol{A} \in \mathbb{R}^{WH \times e}$ and $\widehat{\boldsymbol{A}} \in \mathbb{R}^{wh \times e}$ represent the CSA matrices. They must satisfy the abundance non-negativity constraint (ANC) and the abundance sum-to-one constraint (ASC), ensuring the physical interpretation of abundance as the fraction of material in a pixel (Deng et al., 2025). The deep learning-based unsupervised fusion typically sets two modalities corresponding to LR-HSI and HR-MSI. One modality focuses on obtaining ESD, while another concerns CSAs. We integrate $\boldsymbol{A}$ and $\boldsymbol{E}$ as:

$$\boldsymbol{Z} = \boldsymbol{A}\boldsymbol{E} \tag{2}$$

where $\boldsymbol{Z} \in \mathbb{R}^{WH \times G}$ is the reconstructed HR-HSI. The effectiveness of fusion in Eq. (2) depends on whether the row vectors of $\boldsymbol{A}$ correspond to the column vectors of $\boldsymbol{E}$, representing the same material. This correspondence plays a crucial role in determining the quality of the fused data. Increasing matching information between the two modalities is also the main motivation of this paper.

### 2.2 DEEP FREQUENCY FILTERING

Deep Frequency Filtering (DFF) is a technique introduced to enhance the domain generalization ability of deep neural networks by explicitly modulating the frequency components of learned features (Lin et al., 2023). DFF and Frequency Domain Analysis (FDA) in image processing are closely related, particularly when it comes to manipulating and enhancing features in the frequency domain. For image processing, FDA's basic idea involves transforming an image from the spatial domain to the frequency domain (Wang et al., 2020). DFF shares a common theoretical foundation with traditional FDA and applies this concept to deep learning to tackle domain generalization, improving the model's performance by modulating its learned features in the frequency domain.

Following the research in (Lin et al., 2023), DFF adopts Fast Fourier Transform (FFT) (Cooley & Tukey, 1965) to obtain the feature representations in the frequency domain. Let $\boldsymbol{F} \in \mathbb{R}^{H \times W \times d}$ be the intermediate features, and the Discrete FFT (DFFT) is denoted as:

$$\boldsymbol{F}^{DFFT}(i,j) = \sum_{u=0}^{H-1} \sum_{v=0}^{W-1} \boldsymbol{F}(u,v) \cdot e^{-p2\pi\left(i\frac{u}{H} + j\frac{v}{W}\right)} \tag{3}$$

where $\boldsymbol{F}^{DFFT} \in \mathbb{R}^{2d \times H \times (\lfloor \frac{W}{2} \rfloor + 1)}$ is the frequency representation of $\boldsymbol{F}$, $(i,j)$ denotes the frequency index, $(u,v)$ defines the spatial index, and $p$ is the imaginary unit. $\lfloor \cdot \rfloor$ rounds a number down to the nearest integer less than.

Afterward, the real and imaginary parts of the frequency components are separated for deep feature representation. The FFT decomposes the signal into its frequency components, which are represented as complex numbers. Subsequently, the Inverse DFFT (IDFFT) (Katznelson, 2004) is implemented as:

$$\boldsymbol{F}(u,v) = \frac{1}{HW} \sum_{u=0}^{H-1} \sum_{v=0}^{W-1} \boldsymbol{F}^{DFFT}(i,j) \cdot e^{p2\pi\left(i\frac{u}{H} + j\frac{v}{W}\right)} \tag{4}$$

By cascading multiple layers of processing, DFF can effectively isolate or amplify specific frequency components, offering enhanced performance in applications in signal, image, audio, and video processing (Jiang et al., 2024).

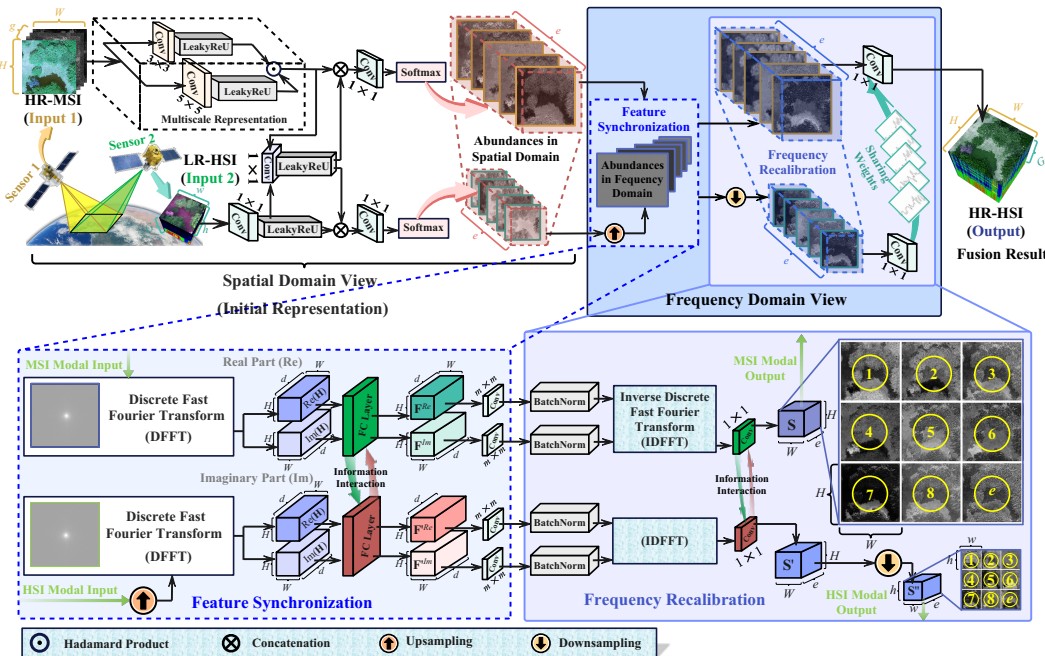

Figure 2: The overview of FSRVL. This work employs the discrete fast Fourier transform to transfer the features from the spatial domain to the frequency domain.

# 3 METHODOLOGY

## 3.1 OVERALL PIPELINE

Figure 2 illustrates the overall pipeline of FSRVL for MSI-HSI fusion. This work employs convolutions to achieve the multiscale representation of local correlations and adjust feature dimension. It adopts the DFFT to transfer the features from the spatial domain to the frequency domain, after aggregating the local correlations. The frequency recalibration in FSRVL will enhance information correspondence. As the unsupervised training progresses, the hidden layers generate CSAs with higher matching values, and their node weights are adjusted accordingly to provide an ESD that better matches the CSAs. Ultimately, the HR-HSI is reconstructed by the ESD and CSAs. *Feature Synchronization* and *Frequency Recalibration* are two key components of innovation. Despite the new method allowing their removal in the framework of MSI-HSI fusion, the models' performance is expected to drop significantly.

## 3.2 INITIAL REPRESENTATION

Given the high spatial resolution of HR-MSI, MSI-HSI fusion relies on it to provide spatial details. To enhance the retention of local relationships between pixels, we combine 2D convolutions and LeakyReLU (Maas et al., 2013) to achieve multiscale representation, which obtains $X_{3\times3} \in \mathbb{R}^{WH\times d}$ and $X_{5\times5} \in \mathbb{R}^{WH\times d}$. LeakyReLU can mitigate the dying neuron issue to ensure training stability. Corresponding to the left side in Figure 2, the multiscale embedding of HR-MSI is denoted as:

$$X' = X_{3\times3} \odot X_{5\times5} \tag{5}$$

where $\odot$ defines the Hadamard Product, $X' \in \mathbb{R}^{WH\times d}$ denotes multiscale features.

Since LR-HSI is utilized to provide ESD, we do not need to pay attention to its spatial information via a multiscale representation. This part utilizes $1 \times 1$ convolutions to adjust feature dimensions. Likewise, we integrate the $1 \times 1$ convolutions and LeakyReLU to implement the LR-HSI embedding $Y' \in \mathbb{R}^{wh\times d}$, and the process is stacked to obtain $Y'' \in \mathbb{R}^{wh\times d}$. Additionally, we utilize $1 \times 1$ convolutions and LeakyReLU to further process $X'$ to acquire $X'' \in \mathbb{R}^{WH\times d}$, which enables consistent modeling of multi-source representation by sharing weights.

Considering that the CSAs reflect the distribution of materials and preserve their spatial information, this work applies Softmax to ensure the ANC and the ASC:

$$\boldsymbol{H} = \text{Softmax}\left(\text{Concat}(\boldsymbol{X}', \boldsymbol{X}'')\boldsymbol{W}_0\right)$$
$$\widehat{\boldsymbol{H}} = \text{Softmax}\left(\text{Concat}(\boldsymbol{Y}', \boldsymbol{Y}'')\boldsymbol{W}_1\right)$$

(6)

where $\boldsymbol{H} \in \mathbb{R}^{WH \times d}$ and $\widehat{\boldsymbol{H}} \in \mathbb{R}^{wh \times d}$ are the initial representation of CSAs, $\boldsymbol{W}_0 \in \mathbb{R}^{2d \times d}$ and $\boldsymbol{W}_1 \in \mathbb{R}^{2d \times d}$ are two learnable weight matrices.

## 3.3 FEATURE SYNCHRONIZATION

We represent each column vector in $\boldsymbol{H}$ or $\widehat{\boldsymbol{H}}$ by hidden layer nodes, and the edge matrices connecting to the next layer are denoted as $\boldsymbol{W}$ or $\widehat{\boldsymbol{W}}$. Each row vector in $\boldsymbol{W}$ or $\widehat{\boldsymbol{W}}$ corresponds to the edge of a node. To prepare for frequency domain operations, we reshape $\boldsymbol{H}$ and $\widehat{\boldsymbol{H}}$ as the tensors $\mathbf{H} \in \mathbb{R}^{W \times H \times d}$ and $\widehat{\mathbf{H}} \in \mathbb{R}^{w \times h \times d}$. Then, we bilinearly interpolate $\widehat{\boldsymbol{H}}$ to match the size of $\boldsymbol{H}$.

Specifically, let $\widehat{\boldsymbol{H}}_c$ be the $c$-$th$ component of $\widehat{\mathbf{H}}$, and $\boldsymbol{H}_c$ is the $c$-$th$ component of $\mathbf{H}$, $c = 1, 2, \ldots, d$. We map the indices $(U, V)$ of $\boldsymbol{H}_c$ to floating coordinates $(u, v)$ in $\widehat{\boldsymbol{H}}_c$, obtaining $u = \frac{U}{W-1} \cdot (w-1)$, $v = \frac{V}{H-1} \cdot (h-1)$. The bilinear interpolation of $\widehat{\boldsymbol{H}}$ is written as:

$$\boldsymbol{H}'_c[U, V, k] = (1 - \delta_u)(1 - \delta_v) \cdot \widehat{\boldsymbol{H}}_c[u_1, v_1, k]$$
$$+ \delta_u(1 - \delta_v) \cdot \widehat{\boldsymbol{H}}_c[u_2, v_1, k]$$
$$+ \delta_v(1 - \delta_u) \cdot \widehat{\boldsymbol{H}}_c[u_1, v_2, k]$$
$$+ \delta_u \delta_v \cdot \widehat{\boldsymbol{H}}_c[u_2, v_2, k]$$

(7)

where $k$ be the interpolated value of the channel, $u_1 = \lfloor u \rfloor$, $u_2 = u_1 + 1$, $v_1 = \lfloor v \rfloor$, $v_2 = v_1 + 1$, $\delta_u = u - u_1$, and $\delta_v = v - v_1$. The tensor $\mathbf{H}' \in \mathbb{R}^{W \times H \times d}$ consists of $d$ matrices using Equ. (7). For convenience, we treat the DFFT in Equ. (3) as a function $\mathcal{F}_{DFFT}(\cdot)$. Our FSRVL applies the DFFT to transfer features from the spatial domain to the frequency domain:

$$[\text{Re}(\mathbf{H}), \text{Im}(\mathbf{H})] = \mathcal{F}_{DFFT}(\mathbf{H})$$
$$[\text{Re}(\mathbf{H}'), \text{Im}(\mathbf{H}')] = \mathcal{F}_{DFFT}(\mathbf{H}')$$

(8)

where $\text{Re}(\mathbf{H})$ and $\text{Re}(\mathbf{H}')$ are the real parts of complex numbers, $\text{Im}(\mathbf{H})$ and $\text{Im}(\mathbf{H}')$ are the corresponding imaginary part. Considering that both the real part and imaginary parts are in Tensor form, to facilitate information interaction across feature channels, we apply Fully Connected (FC) layers to process the real and imaginary parts:

$$\mathbf{F}^{Re} = \text{FC}\left(\text{Re}(\mathbf{H})\right), \quad \mathbf{F}^{Im} = \text{FC}\left(\text{Im}(\mathbf{H})\right)$$

(9)

where $\mathbf{F}^{Re} \in \mathbb{R}^{W \times H \times d}$ and $\mathbf{F}^{Im} \in \mathbb{R}^{W \times H \times d}$ represent the frequency embedding with HR-MSI. Similarly, we also adopt Equ. (9) to process $\text{Re}(\mathbf{H}')$ and $\text{Im}(\mathbf{H}')$ to acquire the frequency embedding with LR-HSI, obtaining $\mathbf{F}'^{Re} \in \mathbb{R}^{W \times H \times d}$ and $\mathbf{F}'^{Im} \in \mathbb{R}^{W \times H \times d}$. The bottom left of Figure 2 provides the schematic of the feature synchronization.

## 3.4 FREQUENCY RECALIBRATION

After conducting the DFFT, if spectral convolution updates a value in the spectral domain, all original nodes will be affected globally (Huang et al., 2025). Therefore, we can still employ 2D convolutions as the learnable filters to capture the global frequency features. In the stream with HR-MSI, the process with the $c$-$th$ channel can be written as:

$$\boldsymbol{Q}_c^{Re} = \text{Con}^{m \times m}(\boldsymbol{F}_c^{Re}), \quad \boldsymbol{Q}_c^{Im} = \text{Con}^{m \times m}(\boldsymbol{F}_c^{Im})$$

(10)

where $\boldsymbol{Q}_c^{Re} \in \mathbb{R}^{W \times H}$ and $\boldsymbol{Q}_c^{Im} \in \mathbb{R}^{W \times H}$ are the Global Frequency Feature (GFF) matrices of the $c$-$th$ channel, $m$ defines the size of the convolution kernel. These GFF matrices with all channels can be combined into the feature tensor of real and imaginary parts: $\mathbf{Q}^{Re} \in \mathbb{R}^{W \times H \times d}$ and $\mathbf{Q}^{Im} \in \mathbb{R}^{W \times H \times d}$.

For the embedding of LR-HSI, we adopt the same strategy in Equ. (10) to process $\mathbf{F}'^{Re}$ and $\mathbf{F}'^{Im}$ for calculating the GFF matrices. Like the above processing, the GFF matrices are integrated as the tensors of real and imaginary parts: $\mathbf{Q}'^{Re} \in \mathbb{R}^{W \times H \times d}$ and $\mathbf{Q}'^{Im} \in \mathbb{R}^{W \times H \times d}$. We employ Batch Normalization (BatchNorm) (Ioffe & Szegedy, 2015) to reduce internal covariate shift, thereby stabilizing the training. BatchNorm will generate Recalibrated Feature Embeddings (RFEs) in the frequencey domain: $\mathbf{B}^{Re} \in \mathbb{R}^{W \times H \times d}$, $\mathbf{B}^{Im} \in \mathbb{R}^{W \times H \times d}$, $\mathbf{B}'^{Re} \in \mathbb{R}^{W \times H \times d}$, and $\mathbf{B}'^{Im} \in \mathbb{R}^{W \times H \times d}$. By the IDFFT in Equ. (4), these features are transformed back to the spatial domain:

$$\begin{aligned} \mathbf{S} &= \mathrm{Con}^{1 \times 1}\left(\mathcal{F}_{IDFFT}[\mathbf{B}^{Re}, \mathbf{B}^{Im}]\right), \\ \mathbf{S}' &= \mathrm{Con}^{1 \times 1}\left(\mathcal{F}_{IDFFT}[\mathbf{B}'^{Re}, \mathbf{B}'^{Im}]\right) \end{aligned} \tag{11}$$

where $\mathbf{S} \in \mathbb{R}^{W \times H \times e}$ and $\mathbf{S}' \in \mathbb{R}^{W \times H \times e}$ are the RFEs in the spatal domain, $\mathcal{F}_{IDFFT}[\cdot, \cdot]$ expresses implementation of the IDFFT. Moreover, we use a bilinear downsample to address $\mathbf{S}'$, thereby restoring the spatial size of LR-HSI to obtain $\mathbf{S}'' \in \mathbb{R}^{w \times h \times e}$. Let each component of $\mathbf{S}$ and $\mathbf{S}''$ correspond to an abundance node, and its weight can represent a vector in the ESD. Consequently, ESDs associated with $\mathbf{S}$ and $\mathbf{S}''$ will also undergo continuous updates as iterations. The bottom right of Figure 2 illustrates the architectures of the frequency recalibration.

## 3.5 IMAGE RECONSTRUCTION

Mathematically, $\widehat{E}$ and $\widehat{A}$ in Equ. (1) can be further decomposed as:

$$\widehat{E} = EP, \quad \widehat{A} = RA \tag{12}$$

where $P \in \mathbb{R}^{G \times g}$ and $R \in \mathbb{R}^{wh \times WH}$ are the blurring factor matrices with the spectral and spatial degradations. Such degradations in the spatial and spectral dimensions are often modeled using the spectral response function and point spread function, respectively. We treat the spectral signals $\mathbf{E}$ and degraded spectral signals $\widehat{\mathbf{E}}$ as learnable weight matrices. According to Equ. (2), we can reconstruct the HR-HSI:

$$Z = \mathrm{Reshape}(\mathbf{S})E, \tag{13}$$

where $\mathrm{Reshape}(\cdot)$ can reshape a tensor to a matrix, $\mathrm{Reshape}(\mathbf{S}) = A$. Following the research in (Liu et al., 2022), if $Z$ exists, $X$ and $Y$ can be deduced in reverse:

$$\widetilde{Y} = \mathrm{Conv}^{\phi \times \psi}(Z), \quad \widetilde{X} = X\widetilde{E} \tag{14}$$

where $\mathrm{Conv}^{\phi \times \psi}$ is a learnable downsampling convolutional layer, ensuring that the downsampled HR-HSI has the same spatial resolution as the LR-HSI. Here, $\phi$ and $\psi$ are two scale factors, $\phi = \lceil H/h \rceil$, $\psi = \lceil W/w \rceil$. The symbol $\lceil \cdot \rceil$ rounds a number up to the nearest integer more than. As the spectral degradation matrix, $\widetilde{E}$ can be computed between the downsampled HR-MSI $X$ and LR-HSI $Y$, and optimized offline prior to model training:

$$\widetilde{E} \Leftarrow \mathrm{argmin}_W \| \mathrm{Downsample}(X) - YW \|_2 . \tag{15}$$

where $W$ is a learnable weight matrix. In FSRVL, the loss function is written as:

$$\begin{aligned} \ell_{\mathrm{FSRVL}} = &\alpha \left( \| Y - \widehat{Y} \| + \| X - \widehat{X} \| \right) \\ &+ \beta \left( \| Y - \widetilde{Y} \| + \| X - \widetilde{X} \| \right) \end{aligned} \tag{16}$$

where $\alpha$ and $\beta$ are the balance coefficients.

## 4 EXPERIMENTS

Table 1 lists five datasets for evaluating fusion performances: the CAVE-Toy, CAVE-Face (Yasuma et al., 2010), TG-1, GF2-GF5, and SZUTree (Long et al., 2024). The CAVE-Toy, CAVE-Face, and TG-1 are used for simulated scenarios. Simulated data provide *Ground Truth (GT)* for quantitative evaluation, but cannot fully reflect real cross-sensor conditions. GF2-GF5 and SZUTree are real cross-sensor datasets. FSRVL employs Adam (Kingma & Ba, 2015) as the optimizer. The environment for implementing FSRVL includes PyTorch 2.4.1, Python 3.9, and CUDA 12.1. The experiments are conducted on a computing server with an NVIDIA RTX 4090 GPU.

Table 1: Dataset types and acquisition platforms.

| TYPE | DATASET NAME | PLATFORM | SOURCE | GROUND TRUTH |
|---|---|---|---|---|
| Registered Data | CAVE-Toy | Ground Camera | Single Sensor | Yes |
| | CAVE-Face | Ground Camera | Single Sensor | Yes |
| | TG-1 | Satellite | Cross-Sensor | Yes |
| Unregistered Data | GF2-GF5 | Satellite | Cross-Sensor | No |
| | SZUTree | UAV | Cross-Sensor | No |

Table 2: Left: *Initial* represents the model without the deep frequency domain filter, *FSRVL-S* represents the model without Feature Synchronization, and *FSRVL-R* represents the model without Feature Recalibration. Right: Features of spatial and frequency domain views

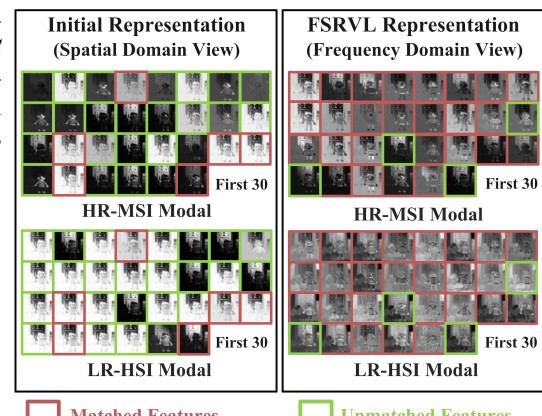

| | *Initial* | *FSRVL-S* | *FSRVL-R* | *FSRVL* |
|---|---|---|---|---|
| SAM↓ | 0.94 | 0.69 | 0.71 | **0.64** |
| PSNR↑ | 45.04 | 48.05 | 47.64 | **48.35** |
| ERGAS↓ | 0.22 | 0.18 | 0.17 | **0.15** |
| SSIM↑ | 0.98 | 0.98 | **0.99** | **0.99** |
| Para (M) | 0.98 | 0.98 | 0.99 | 0.99 |
| Times (H) | 0.19 | 0.50 | 0.59 | 0.63 |

- **CAVE Datasets**: The CAVE-Toy ($512 \times 512 \times 31$) was downsampled into the HR-MSI ($512 \times 512 \times 3$) and the LR-HSI ($64 \times 64 \times 31$), and the CAVE-Face follows the same process.

- **TG-1 Dataset**: The original data ($240 \times 240 \times 54$) was from TianGong01 satellite and downsampled into the HR-MSI ($240 \times 240 \times 8$) and the LR-HSI ($30 \times 30 \times 54$).

- **GF2-GF5 dataset**: GaoFen02 satellite provides the GF2 data-HR-MSI ($360 \times 360 \times 4$), while GaoFen05 satellite collects the LR-HSI ($60 \times 60 \times 330$).

- **SZUTree Dataset**: It was collected by an UAV with a RGB camera and a SPECIM FX10 sensor, yields the HR-MSI ($300 \times 300 \times 3$) and the LR-HSI ($150 \times 150 \times 98$).

Fusion performance is evaluated using five standard metrics: *Spectral Angle Mapper* (*SAM* ↓), *Peak Signal-to-Noise Ratio* (*PSNR* ↑), *Erreur Relative Globale Adimensionnelle de Synthese* (*ERGAS* ↓), *Structural Similarity Index* (*SSIM* ↑), *Mean Relative Absolute Error* (*MRAE* ↓). Here, ↑ indicates that higher values denote better results, while ↓ signifies lower values are preferable.

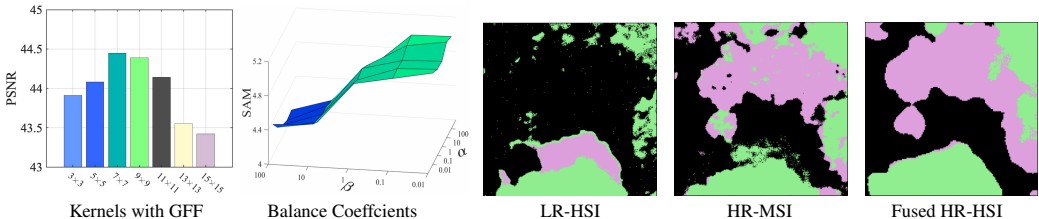

Figure 3: Left compares performances when configuring different hyperparameters. Right lists the classification maps on the SZTree data using FSRVL.

## 4.1 ABLATION STUDY AND HYPERPARAMETER ANALYSIS

Table 2 provides the ablation study on the TG-1 dataset that evaluates the contribution of FSRVL's three modules: Initialization, Feature Synchronization, and Frequency Recalibration. As shown in Table 2, each module plays a significant role in improving fusion performance. Figure 3 illustrates the impact of configuring different hyperparameters on fusion performance. Tests on the CAVE-Face dataset indicate that the convolution kernel in Equ. (10) is $7 \times 7$, with $\alpha = 100$ and $\beta = 10$ yielding the best results, and this setting is used throughout all experiments. Figure 3 also illustrates the classification maps of the SZUTree dataset generated by FSRVL, demonstrating that the fused HR-HSI can facilitate more accurate land-cover classification.

## 4.2 COMPARISON WITH BASELINES

FSRVL is compared with baselines, such as CUCaNet (Yao et al., 2020), SSG-SRL (Liu et al., 2025), DFMFf (Guo et al., 2023), CS2DIPs (Fang et al., 2024), DTDNML (Wang et al., 2025), EDIP (Li et al., 2025), and OTIAS (Deng et al., 2025). Therein, CUCaNet, CS2DIPs, DTDNML, and EDIP are unsupervised approaches, whereas SSG-SRL, DFMFf, and OTIAS are supervised methods.

Figure 4 compares FSRVL, EDIP, and OTIAS on the CAVE dataset, showing that FSRVL yields spectral results closer to the GTs. Figure 5 and Table 3 show that FSRVL outperforms baselines on the TG-1 dataset. The CAVE and TG-1 datasets are both simulated with ideal conditions where data across different modalities are perfectly aligned, enabling unsupervised methods to leverage their full potential. In contrast, supervised methods exhibit slightly inferior fusion performance due to their reliance on sampling.

CS2DIPs, DTDNML, and EDIP are excluded from tests on GF2-GF5 and SZUTree because their implementations require strictly aligned data. Figure 6 shows that FSRVL best preserves the spectral/color features of LR-HSI and the spatial details of HR-MSI. We employ a simple classifier with a fully connected layer followed by a Softmax in the downstream.

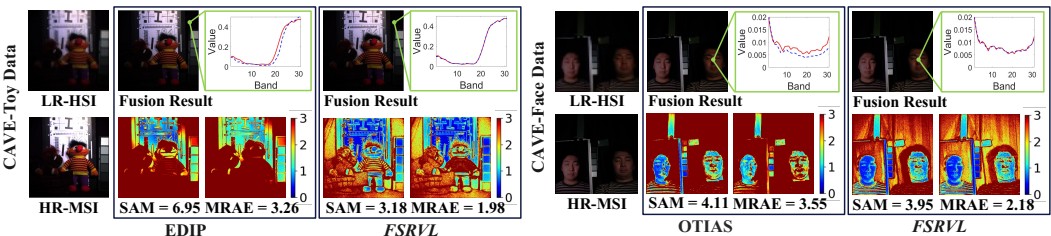

Figure 4: Comparing fusion results on the CAVE datasets, where the red curves represent the Ground Truth, and the blue dotted lines represent the estimated spectra.

Table 3: Quantitative evaluation of different methods on the TG-1 dataset. The **best** and second-best values are highlighted.

| | Method | CUCaNet | SSG-SRL | DFMFf | CS2DIPs | DTDNML | EDIP | OTIAS | *FSRVL* |
|---|---|---|---|---|---|---|---|---|---|
| | Published Year | 2020 | 2025 | 2023 | 2024 | 2025 | 2025 | 2025 | |
| | SAM↓ | 1.21 | 1.14 | 2.23 | 1.15 | 0.99 | 0.91 | 0.89 | **0.64** |
| | PSNR↑ | 40.35 | 44.90 | 44.40 | 44.08 | 44.47 | 45.45 | 45.71 | **48.35** |
| TG-1 | ERGAS↓ | 0.40 | 0.29 | 0.51 | 0.29 | 0.27 | 1.44 | 0.24 | **0.15** |
| | SSIM↑ | 0.95 | 0.96 | 0.95 | 0.96 | 0.97 | 0.97 | 0.97 | **0.99** |
| | Para (M) | 1.78 | 3.36 | 4.82 | 0.30 | 0.71 | 12.25 | 2.14 | 0.99 |
| | Times (H) | 0.54 | 0.93 | 0.87 | 0.10 | 0.39 | 0.61 | 0.66 | 0.63 |

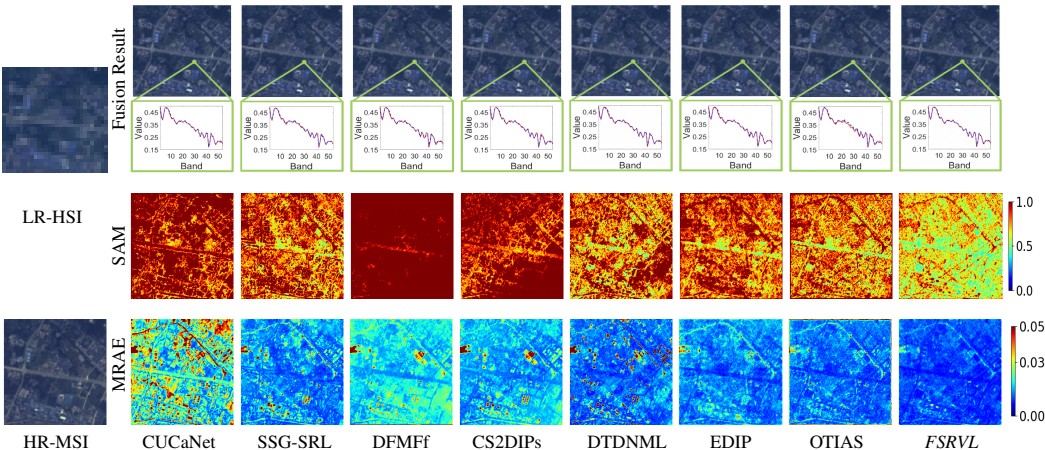

Figure 5: Comparing fusion results on the TG-1 dataset. The top row displays the RGB compositions (R:28, G:18, B:5) of the fused HR-HSIs and the pixels' spectra. The second and third rows present the SAM heat map and the MRAE heat map.

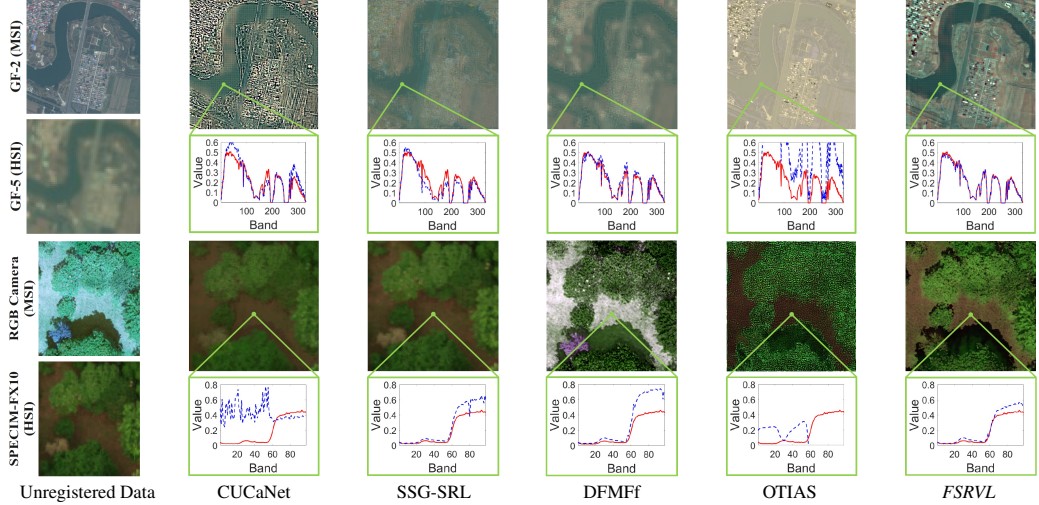

Figure 6: Reconstructed HR-HSIs were obtained by different methods using the unregistered datasets. The two datasets captured the same area within a short time. GTs of spectra are from the LR-HSI.

## 5 CONCLUSION

This work presents our FSRVL, a novel framework for cross-sensor fusion of multispectral and hyperspectral images. We designed the Feature Synchronization module and the Frequency Recalibration module to adaptively modulate frequency features, effectively mitigating spectral-spatial information mismatches between HR-MSI and LR-HSI. Comprehensive experiments on both simulated and real-world datasets validate its robustness and demonstrate superior performance compared to existing methods in spectral image fusion. In future work, we plan to extend FSRVL to handle weakly aligned data and explore its applications in real-time onboard processing. Additionally, I acknowledge that this paper has been revised and polished with the assistance of ChatGPT to improve its clarity and readability. Upon acceptance of the paper, we will make the experimental datasets and demos available for download to ensure reproducibility of the study. Further details are shown in Appendix.

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
