# A APPENDIX

In the supplemental material:

- A.1: We provide the pseudocode of FSRVL.
- A.2: We describe the operation environment of the experiments.
- A.3: We provide additional experimental results obtained using the CAVE-Toy and CAVE-Face datasets.
- A.4: We demonstrate the experimental details of spectral image classification, carried out after applying our FSRVL to perform cross-sensor spectral image fusion on the SZUTree dataset.

## A.1 PSEUDOCODE

The pseudocode for implementing FSRVL is provided in Algorithm 1, which facilitates readers' better understanding of its execution logic.

---

**Algorithm 1:** Pseudocode of FSRVL

---

**Input:** HR-MSI $\mathbf{X}$ and LR-HSI $\mathbf{Y}$.
**Output:** HR-HSI $\mathbf{Z}$.

1 $\mathbf{H}$ and $\widehat{\mathbf{H}} \leftarrow$ Use convolutions, Equ. (5), and Equ. (6) to process $\mathbf{X}$ and $\mathbf{Y}$.
2 **for** *Epochs* **do**
3   **repeat**
5     $\mathbf{H}' \leftarrow$ Employ Equ. (7) to implement the upsample.
7     $\mathbf{F}^{Re}, \mathbf{F}^{Im}, \mathbf{F}'^{Re}$, and, $\mathbf{F}'^{Im} \leftarrow$ Utilize Equ. (3), Equ. (8), and Equ. (9) to achieve the deep discrete fast Fourier transform.
9     $\mathbf{Q}^{Re}, \mathbf{Q}^{Im}, \mathbf{Q}'^{Re}$, and $\mathbf{Q}'^{Im} \leftarrow$ Employ Equ. (10) to calculate the global frequency features.
11     $\mathbf{S}$ and $\mathbf{S}'' \leftarrow$ Use Equ. (4), Equ. (11), and the bilinear downsample for the inverse Fourier transform.
13     $\widetilde{\mathbf{E}} \leftarrow$ Employ the Equ. (15) to estimate the spectral degradation matrix.
15     $\widetilde{\mathbf{X}}$ and $\widetilde{\mathbf{Y}} \leftarrow$ Adopt Equ. (14) to acquire the degradations of the HR-HSI.
17     $\ell_{\text{FSRVL}} \leftarrow$ Calculate the training loss of FSRVL using Equ. (16).
19     $\mathbf{Z} \leftarrow$ Use Equ. (13) to construct the HR-HSI as the output of FSRVL.
20   **until** *Convergence of* $\ell_{\text{FSRVL}}$;
21 **end**

---

## A.2 OPERATION ENVIRONMENT

All experiments were conducted on a high-performance computing server equipped with an NVIDIA GeForce RTX 4090 GPU. The model was trained for $1 \times 10^4$ iterations, using a dynamic learning rate schedule. The initial learning rate of $3 \times 10^{-3}$ was maintained for the first $2 \times 10^3$ iterations, followed by a linear decay to zero over the subsequent $8 \times 10^3$ iterations.

## A.3 EXPERIMENTAL RESULTS WITH THE CAVE DATA

For simulations, we utilized the CAVE data and the TG-1 data collected by TianGong01. The CAVE and TianGong-01 datasets are publicly available and can be accessed upon request from their respective official websites. The CAVE dataset includes 32 high-quality hyperspectral images, each with a spatial resolution of $512 \times 512$ pixels. These images cover 31 spectral bands spanning the visible spectrum from 400 $nm$ to 700 $nm$, with a precise spectral resolution of 10 nm per band. From this dataset, we selected two images, "Face and Photo" and "Toy and Chart," for our subsequent experiments, referred to as CAVE-Face and CAVE-Toy.

Figure 7 illustrates the fusion results with the CAVE-Toy dataset. While all MSI-HSI fusion methods produce sufficiently clear fused images, the heatmap visualizations in the second and third rows

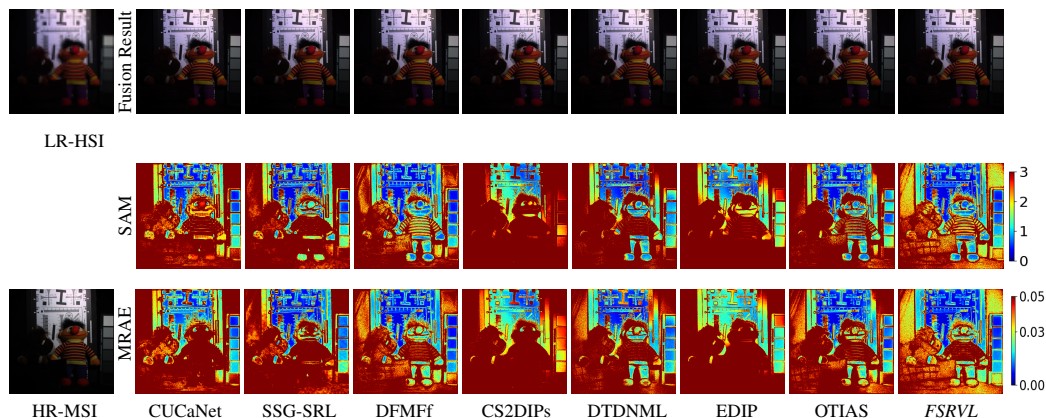

Figure 7: Comparing fusion results on the CAVE-Toy dataset. The top row displays the RGB composition (R:28, G:18, B:5). The second and third rows present the SAM heat map and the MRAE heat map.

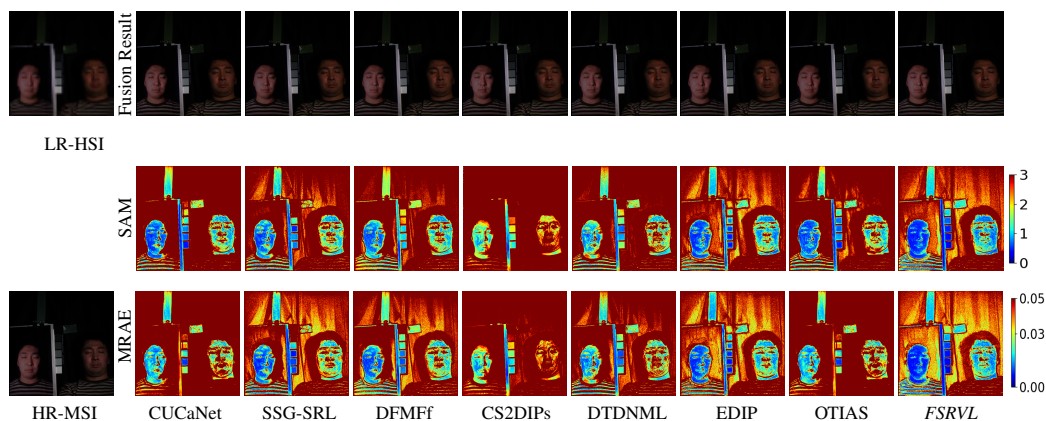

Figure 8: Comparing fusion results on the CAVE-Face dataset. The top row displays the RGB composition (R:28, G:18, B:5).

Table 4: Quantitative comparisons on the CAVE-Toy and CAVE-Face datasets. The **best** and second-best values are highlighted.

|  | Method | CUCaNet | SSG-SRL | DFMFf | CS2DIPs | DTDNML | EDIP | OTIAS | *FSRVL* |
|---|---|---|---|---|---|---|---|---|---|
| CAVE-Toy | SAM↓ | 5.34 | 4.73 | 6.94 | 8.13 | 4.64 | 6.95 | 5.52 | **3.18** |
|  | PSNR↑ | 40.45 | 42.10 | 39.44 | 39.91 | 42.10 | 42.45 | 43.27 | **46.01** |
|  | ERGAS↓ | 0.98 | 0.55 | 1.12 | 1.33 | 0.91 | 1.02 | 0.88 | **0.53** |
|  | SSIM↑ | 0.98 | 0.98 | 0.97 | 0.97 | 0.97 | 0.98 | 0.98 | **0.99** |
| CAVE-Face | SAM↓ | 8.01 | 6.67 | 6.22 | 8.23 | 6.27 | 4.55 | 4.11 | **3.95** |
|  | PSNR↑ | 39.96 | 41.41 | 43.23 | 41.32 | 42.27 | 43.65 | 44.32 | **44.39** |
|  | ERGAS↓ | 2.23 | 1.99 | 1.62 | 2.24 | 1.94 | 1.43 | 1.44 | **1.42** |
|  | SSIM↑ | 0.97 | 0.97 | 0.97 | 0.97 | 0.98 | 0.98 | **0.99** | **0.99** |

reveal significant performance differences. The red regions in the heatmaps represent error magnitudes, with larger red areas indicating higher inaccuracies. Figure 8 shows the fusion results of different approaches on the CAVE-Face dataset. The CAVE-Face dataset experiments were conducted under the same imaging conditions as CAVE-Toy.

Table 4 provides the quantitative comparisons on the CAVE-Toy and CAVE-Face datasets. In the CAVE-Toy side of Table 4, our FSRVL demonstrates superior spectral preservation, as evidenced by its leading SAM scores. Meanwhile, the proposed method attains the highest PSNR values. In the CAVE-Face side of Table 4, our FSRVL also exhibits advantages over conventional deep learning-based fusion methods.

## A.4 EXPERIMENTAL DETAILS OF SPECTRAL IMAGE CLASSIFICATION

In real cross-sensor scenarios, we obtained GF2 and GF5 data from the GaoFen satellite constellation, and RGB (it is regarded as multispectral data) and hyperspectral data from the SZUTreeData 2.0 Multimodal dataset published by Shenzhen University. GF2 is a multispectral satellite, while GF5 is a hyperspectral satellite. These satellites are capable of capturing the same ground scene within a short time interval, providing truly cross-sensor satellite multispectral and hyperspectral imagery for our experiments. Meanwhile, the SZUTreeData 2.0 dataset offers cross-sensor UAV multispectral and hyperspectral images, captured using an RGB camera and a SPECIM FX10 hyperspectral sensor. GF2 and GF5 were used to conduct spaceborne experiments, while the SZUTreeData was used for airborne experiments. As a result, the experiments in our manuscript comprehensively cover scene testing across ground-based, airborne, and spaceborne platforms.

The GF2-GF5 and SZUTree datasets used in our study, along with the source code, will be packaged into a runnable demo and made available on our website after the successful publication of the paper. Since the experiments using GF2-GF5 and SZUTree data play a critical role in validating the effectiveness of FSRVL, we included both in the main text. Additionally, the SZUTree dataset provides land cover category information for the imaged area. We manually interpreted and labeled regions of interest (ROIs) based on this information to serve as ground truth (GT) for the classification task. In the experiment using the SZUTree dataset, we randomly selected 15% of the samples from each class as the training set, and the remaining samples were used as the test set. The fused image shares the ROIs with the HR-MSI.

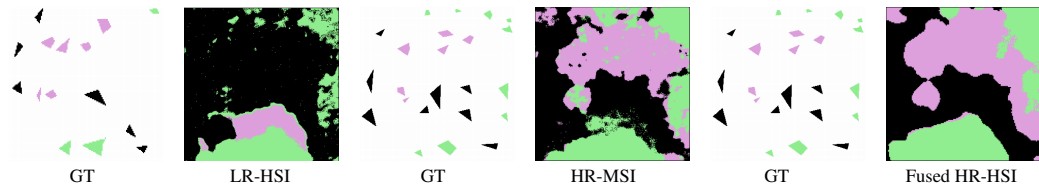

| GT | LR-HSI | GT | HR-MSI | GT | Fused HR-HSI |

Figure 9: Classifications on the SZTree data using our FSRVL.

Table 5: Comparison of accuracies on the SZTree dataset. Evaluation metrics: OA, AA, and Kappa coefficient ($\kappa$).

|  |  | OA↑ | AA↑ | $\kappa$↑ |
|---|---|---|---|---|
|  | LR-HSI | 64.54% | 57.11% | 0.4460 |
| SZTree Data | HR-MSI | 82.74% | 88.70% | 0.8456 |
|  | HR-HSI | 90.62% | 91.84% | 0.8853 |

Figure 9 shows the classification maps by FSRVL, utilizing the LR-HSI, HR-MSI, and Fused HR-HSI. Table 5 provides the classification accuracy results for LR-HSI, HR-MSI, and Fused HR-HSI. The performance metrics used for evaluation include Overall Accuracy (OA), Average Accuracy (AA), and Kappa coefficient ($\kappa$), which are commonly employed to assess the effectiveness and reliability of the classification results. The results in Figure 9 and Table 5 demonstrate that the HSI fused by FSRVL makes it easier to achieve high-precision pixel-level classification.