# OpenReview forum: "Unsupervised Multi-Sensor Spectral Image Fusion via Frequency-Spatial Reciprocal-View Learning"
_ICLR.cc/2026/Conference — ICLR 2026 Conference Withdrawn Submission_

### Official Review · Reviewer_Xaa2 · 2025-10-28

**Soundness:** 3
**Presentation:** 3
**Contribution:** 2
**Rating:** 4
**Confidence:** 3

**Summary:**

This paper proposes FSRVL (Frequency–Spatial Reciprocal-View Learning) for unsupervised fusion of multispectral (MSI) and hyperspectral (HSI) images. The key idea is to address cross-sensor misalignment by jointly processing MSI and HSI features in the frequency domain. The model includes two major modules: Feature Synchronization (FS) and Frequency Recalibration (FR). Experiments on both simulated (CAVE-Toy, CAVE-Face, TG-1) and real unregistered datasets (GF2–GF5, SZUTree) show improved SAM, PSNR, and SSIM compared with recent baselines such as EDIP, DTDNML, and OTIAS.

**Strengths:**

+ The paper correctly identifies the critical issue of non-perfect registration in MSI–HSI fusion and attempts to address it via frequency-domain synchronization.
+ Introducing DFFT-based joint representation and recalibration is conceptually interesting, showing awareness of recent works on frequency filtering for domain generalization.
+ Evaluations on both aligned and unaligned datasets and inclusion of ablations lend empirical completeness.
+ The paper is well written, with clear figures and an understandable flow.

**Weaknesses:**

- The frequency recalibration is empirically motivated, with little explanation of why frequency features better handle spatial misalignment. No mathematical or intuitive link is drawn between Fourier feature coupling and physical spectral correspondence.
- Why should frequency filtering inherently alleviate spatial misalignment? Without a geometric correction model, frequency modulation alone might not guarantee correspondence.
- Comparisons are restricted to a subset of prior works. Some strong baselines are missing (e.g., UAL: Unsupervised Adaptation Learning for Hyperspectral Imagery Super-resolution, DAEM: Toward Stable, Interpretable, and Lightweight Hyperspectral Super-resolution).
- The experimental validation on the CAVE dataset relies on a small number of selected samples rather than the full dataset. This raises concerns about the robustness and generalizability of the reported performance.
- The error visualization is presented within a very narrow value range, making it difficult to accurately assess the performance advantages of FSRVL over other methods.

**Questions:**

Refer to the Weaknesses.

---

### Official Review · Reviewer_XU76 · 2025-10-31

**Soundness:** 2
**Presentation:** 3
**Contribution:** 2
**Rating:** 2
**Confidence:** 5

**Summary:**

The paper proposes Frequency–Spatial Reciprocal-View Learning (FSRVL) for unsupervised MSI–HSI fusion under cross-sensor mis-registration. It shares weights to “synchronize” MSI/HSI features, moves abundances to the frequency domain via (I)DFFT with FC/conv blocks for frequency recalibration, and reconstructs HR-HSI by combining learned abundances and endmembers. Experiments on CAVE, TG-1 (simulated), and GF2-GF5, SZUTree (unregistered) report gains over several baselines; ablations suggest both synchronization and recalibration help.

**Strengths:**

Originality: Applies deep frequency filtering to MSI–HSI fusion and explicitly targets cross-sensor mismatch.

Quality: Includes ablations; uses both simulated and real unregistered data; reports multiple metrics.

Clarity: High-level intuition and pipeline figure are provided; contributions are enumerated.

Significance: Addresses a relevant pain point (imperfect registration) in unsupervised fusion.

**Weaknesses:**

Novelty is incremental. Weight sharing + FFT/IFFT + conv gating is a fairly direct adaptation of DFF; no principled link between frequency manipulation and spatial mis-registration is established (no analysis of tolerance vs. pixel shift/PSF/SRF mismatch).

Modeling gaps. LSMM is assumed, yet ASC/ANC are enforced via Softmax on hidden activations without verifying endmember interpretability; spectral/spatial degradations P, R are treated loosely (Eq. 12–16) with unclear identification and limited physical grounding.

No controlled mis-registration study (e.g., varying sub-pixel shifts/rotations/PSF) to substantiate the core claim.

Several unsupervised baselines are excluded from unregistered sets because they need alignment, weakening the fairness of comparisons precisely where the paper claims an advantage.

Reproducibility. Code “will be linked,” datasets/demos released only upon acceptance; insufficient training details (data splits, preprocessing, SRF/PSF handling, runtime/memory, seeds).

Notation/technical issues. FFT equations mix symbols (e.g., p as imaginary unit), index reuse, and minor inconsistencies around shapes; Eq. (14–15) downsampling/spectral degradation learning is under-specified; figure captions sometimes over-claim.

**Questions:**

Can you provide systematic mis-registration experiments (synthetic shifts/rotations/blur, varying SRF/PSF) with curves of SAM/ERGAS vs. misalignment magnitude?

How do Softmax-enforced abundances retain ASC/ANC while preserving material semantics across modalities? Any endmember purity/orthogonality checks?

What is the role of frequency recalibration vs. simple spatial attention/deformable conv? Please compare to deformable kernels and cross-attention with offset learning under the same setting.

How are P, R, E estimated in practice? Are SRFs known/approximated? Please detail training losses, initializations, and constraints.

Provide full reproducibility: code, scripts, SRF/PSF configs, seeds, and protocols; include baseline re-tuning on your data.

---

### Official Review · Reviewer_s6Aa · 2025-11-01

**Soundness:** 2
**Presentation:** 3
**Contribution:** 4
**Rating:** 6
**Confidence:** 3

**Summary:**

The paper proposes FSRVL, a frequency-domain framework for unsupervised fusion of low-resolution hyperspectral images (LR-HSI) and high-resolution multispectral images (HR-MSI) under non-absolute registration. It introduces the Feature Synchronization module and the Frequency Recalibration module to adaptively modulate frequency features, effectively mitigating spectral-spatial mismatches between HR-MSI and LR-HSI. The method demonstrates superior performance over existing baselines on both simulated and real-world datasets.

**Strengths:**

The paper addresses MS-HSI fusion under non-absolute registration with an original frequency-domain framework that integrates a Feature Synchronization path and a Frequency Recalibration path to alleviate spectral-spatial mismatch. The methodology is well-motivated . The overall pipeline is clearly structured and easy to follow. Although the experimental coverage is not exhaustive, the reported results on both simulated and selected real-world scenarios are encouraging and demonstrate the method’s potential effectiveness.

**Weaknesses:**

The experimental coverage is too narrow to support broad claims. CAVE contains many distinct scenes, yet the paper reports only two, this makes conclusions about the learned ESD/CSAs far less persuasive. In addition, most CAVE results appear to rely on HR-MSI that is synthesized from HSI (near-perfect registration), which naturally favors learning ESD from LR-HSI and CSAs from HR-MSI and therefore does not truly stress the proposed handling of non-absolute registration. Robustness is also unclear, there is no controlled study on misregistration magnitude (translation/rotation/scale), even though this is the paper’ s core motivation.

**Questions:**

1.The formulation (14) X=XE appears questionable. It introduces E before it is defined, and more importantly, it contradicts the underlying physical generation process of MSI from HSI. Please double-check whether this equation is valid and consistent with the rest of the framework, especially in terms of tensor shapes and modality alignment.
2.The method is designed for non-absolute registration, yet no controlled experiments are conducted to assess robustness against misalignment. Could you synthesize artificial misregistration (e.g., translation, rotation, scale) on TG-1 and a broader CAVE subset—not just two scenes—and report performance trends (e.g., PSNR/SAM) with respect to misalignment magnitude? A weak-alignment baseline would also help contextualize the claimed improvements.
3.The 7×7 kernel and loss weights (α=100, β=10) seem to be selected on the CAVE–Face scene and then directly applied to all experiments. This raises concerns about potential overfitting or lack of generality. If these values are kept fixed, a brief explanation of their transferability—ideally with confidence intervals or performance variance across datasets—would be helpful.
4.The conclusion states that code and data will be released “upon acceptance”, which makes it difficult to verify results during review. If sharing the full source is not feasible at this stage, consider providing a black-box runner with fixed seeds, full configuration files, and precomputed reconstructions. At minimum, please release the exact data splits, complete training/inference logs, and links or specifications for the datasets used.

---

### Note · Authors · 2025-12-22

I have read and agree with the venue's withdrawal policy on behalf of myself and my co-authors.